# CALVIN: Improved Contextual Video Captioning via Instruction Tuning

**Gowthami Somepalli**⋆♥, **Arkabandhu Chowdhury** ♠,

**Ronen Basri** ♣, **Jonas Geiping** ♦, **Tom Goldstein** ♥, **David Jacobs**♥

♥ University of Maryland, College Park       ♠ Meta AI

♦ ELLIS Institute, MPI for Intelligent Systems       ♣ Weizmann Institute of Science

## Abstract

The recent emergence of powerful Vision-Language models (VLMs) has significantly improved image captioning. Some of these models are extended to caption videos as well. However, their capabilities to understand complex scenes are limited, and the descriptions they provide for scenes tend to be overly verbose and focused on the superficial appearance of objects. Scene descriptions, especially in movies, require a deeper contextual understanding unlike general-purpose video captioning. To address this challenge, we propose a model, CALVIN, a specialized video LLM that leverages previous movie context to generate fully "contextual" scene descriptions. To achieve this, we train our model on a suite of tasks that integrate both image-based question-answering and video captioning within a unified framework, before applying instruction tuning to refine the model's ability to provide scene captions. Lastly, we observe that our model responds well to prompt engineering and few-shot in-context learning techniques, enabling the user to adapt it to any new movie with very little additional annotation.

---

**Ground Truth:** Lenihan knocks it backwards into the pool

**VideoLlama:** First, the video shows a soldier holding a rifle and standing on stairs. Next, a soldier holding a gun is seen standing on a rooftop in the rain.....the soldier is holding a bow and arrow and a rifle.

**VideoLlaVa:** The video shows a man in a military uniform holding a rifle and aiming it at a target. He then fires the rifle.Here are several guns visible in the image, including a rifle held by a soldier, a handgun, and a few other guns scattered around the area.

**MovieChat:** The video shows various scenes of an army man in a military setting, including a helicopter flying, a soldier standing in front of a building, a soldier standing in front of a tree, and soldiers walking

**CALVIN (Ours):** Lenihan shoots the alien in the head

---

Figure 1: A scene from MAD [80]-eval split. We present the captions generated by our model, represented as CALVIN against various off-the-shelf LLMs, with hallucinations highlighted in red. First, our model utilizes the context well, by understanding the name of the character is 'Lenihan' and that there is an alien in the scene, and second, our model has less hallucination and verbosity compared to other models.

---

⋆ Work done during internship at Meta. Correspondence: gowthami@umd.edu

38th Conference on Neural Information Processing Systems (NeurIPS 2024).

# 1   Introduction

The volume of video data on the internet is increasing every year, now representing the largest portion of internet traffic [2]. To make this wealth of visual content accessible to vision impaired individuals requires audio descriptions. Audio descriptions (AD) describe and narrate videos or segments of videos in natural language, but are often still manually curated for only a few select videos.

The emergence of large, multi-modal vision-language models (vLLMs) has led to exciting progress on supplementing AD with automatically generated descriptions (AAD), but current models still suffer from notable weaknesses in videos. Consider the concrete example of a video of a woman smiling and waving at a bus carrying a friend. An observer with understanding of the context of the scene, like a human, might caption it as `A happy woman waving at the bus, perhaps receiving a loved one`, while a vLLM trained on image-caption pairs would provide a more literal description, such as `A woman wearing a red dress. A woman is smiling. A bus stopped. The scene has people in the background`. This illustrates a key shortfall: vision LLMs focus on superficial details like the properties of physical objects, which are often irrelevant to the broader context, because they are trained on static images. Several of these models process videos only as individual images, simply stringing together descriptions from each image while failing to capture the overarching narrative.

Secondly, current systems struggle to incorporate prior context into their description. Using the same scenario, if we inform a human that the woman's name is Mary and her husband, a soldier, is returning from war, a human would caption it as, `Mary is overflowing with joy to receive her husband, a war hero, at the bus stop`. Here, the human might choose to omit less significant details like 'waving' in favor of emphasizing her happiness, showing a nuanced understanding of the context. In contrast, many vision LLMs lack the ability to integrate such context with visual data, often ignoring it entirely. This is a significant gap, as useful interpretations naturally prioritize the intentions of characters and the results of their interactions through time, rather than the mere presence of objects and their movements - which would make the video hard to follow for a listener reliant on AD.

Drawing inspiration from the way humans utilize context in captioning, we introduce a novel contextual captioning model, CALVIN, that is designed and instruction-tuned to generate audio descriptions. Our primary objective is to develop a model that, given appropriate context, can generate captions closely resembling those crafted by humans. To accomplish this, we train our model using the Movie Audio Descriptions(MAD) dataset [80], which includes human-generated annotations for movie scenes, complete with timestamps. This enables us to construct a "text context" for each scene, based on preceding scenes. However, the MAD dataset alone is limited in scope and insufficient for fully training a video-LLM. To address this, we incorporate image-VQA datasets, which significantly enhance the model's visual understanding. We provide a detailed discussion on training methodologies, including the optimal combinations of data to use at different stages.

In addition to producing scene descriptions that are more human-like and useful than the existing vision models, CALVIN, achieves state-of-the-art (SOTA) performance in captioning on the MAD-eval dataset, with major improvements ($\sim$26% improvement on CIDEr and $\sim$68% improvement on BertScore) over the previously established SOTA model. CALVIN also performs significantly better than all the recent off-the-shelf video LLMs we studied on zero-shot evaluations on the TVC dataset [43]. We illustrate the model's capabilities in Figure 1. Here, while most video-LLMs generically describe the scene as `a soldier holding a rifle` or `an army man in a military setting` missing the narrative nuance of the ground truth caption `Lenihan knocks it backward into the pool`. CALVIN stands out by not only recognizing the character as 'Lenihan' but also incorporating prior information about the presence of an alien. Unlike the overly verbose captions of other models, our captions are crisp and narrative, focusing on actions and outcomes that are important to the plot, underlining their usefulness for the task of automated audio description.

We also introduce two test-time adaptation strategies, prompting and few-shot tuning, that are particularly useful in scenarios where additional contextual information is lacking. Finally, we describe some of the limits of our current model and outline potential avenues for future research to further advance this field.

## 2 Related Work

**Video Understanding.** The key objective of parsing spatiotemporal information in videos can be achieved through hand-crafted features [15, 18, 22, 69], recurrent networks [19, 36, 107], convolutional networks [24, 27, 51, 70, 90, 95], and more recently Vision Transformers (ViTs) [7, 9, 21, 23]. ViTs [21] treat an image as a set of patches and use a transformer architecture to model their interactions. Some works also effectively add multi-scale hierarchies [23, 28, 60, 73, 106] or local structures [14, 20, 60] to the transformers. Naturally these models can be extended to videos where a video is treated as a sequence of independent image frames, and a subsequent temporal pooling layer or a temporal transformer is added [7, 16]. There have been more generalized video modeling approaches [9, 23, 66, 68, 73, 84] that directly work on a video clip by dividing it into 3D spatio-temporal patches. While most of these models have proven to work well on short videos (<5 seconds), longer-video (>30 seconds) understanding is still an active research area. Some existing methods include pre-computing features and separately training backbones [3, 19, 26, 94], increasing model efficiency to include more frames [35, 38, 95, 110], and building a memory-model that can reference prior context [13, 41, 42, 96, 97]. Recent studies show that video-text pretraining [4, 5, 25, 29, 37, 45, 46, 49, 63, 71, 75, 82, 88, 89, 101, 103] can greatly help in long video understanding [83], temporal localization tasks [12, 44, 55, 100], text-video retrieval [63], video question answering [104], and video clip captioning [5, 75].

**Generalist Video LLMs.** Recent breakthroughs in language modeling have also spurred a flurry of work incorporating first image data, and then video data, as additional input modalities [5, 61]. While some works try to train large-scale video-text models directly from scratch [47, 65, 67, 87], most work focuses on continued pretraining and finetuning of base language models [102, 108]. Models may be generic multi-modal models [98, 105], or branch off from existing image-text models, such as LLaVA [52], targeting video understanding [54], and initial work on long video understanding [54]. Instruction tuning for videos was considered in [58, 59] and [78], and, with a particular emphasis on interactions with long(er) videos in [48, 81]. Understanding long videos is not only a learning problem, but also a technical challenge, necessitating engineering improvements, such as RingAttention to even process long videos [57].

**Contextual Video Captioning.** What are use cases for these video LLMs? A particularly interesting one is automated video captioning, which effectively converts video content back into text. This is useful, not only for tasks such as search, retrieval, but also essential to generate audio descriptions (AD) of video content. Automated audio descriptions describe the content of videos verbally, and are central to making videos accessible to anyone with visual impairment [64, 76, 93]. Seminal work, such as CineAD [11], argued that automated audio descriptions of even highly contextual content such as cinema, should be feasible, leading to a broader interest in automated audio descriptions (AAD). The most recent work in this direction are AutoAD I and II [30, 31], generating audio descriptons based on CLIP features processed with smaller language models. Auto-AD III [32] is a concurrent work that introduces a large-scale dataset and a 7b model to perform this task. AD is particularly focused on contextual descriptions and allows a listener to make sense of a long-form video, such as a movie. This separates this task from the more generic task of dense video captioning [33, 39, 77, 82, 91].

## 3 CALVIN: A contextual video captioner

We now discuss the proposed architecture, datasets, and training process.

### 3.1 Method & Architecture

Consider a collection, $\mathcal{V}$, of movie scenes $[\mathbf{v_1}, \mathbf{v_2}, \ldots, \mathbf{v_n}]$. Each scene contains $k$ image frames such that $\mathbf{v_j} = [v_j^1, \ldots, v_j^k]$, and comes with annotations $\mathbf{v}_j$ in the form of a sequence of text tokens $[a_j^1, a_j^2, .., a_j^m]$. These annotations are usually obtained by converting audio to text, where the audio comes from the movie dialog or a person describing the scene. We seek a model $f_\phi(\cdot)$ that effectively predicts the audio description given the past visual and text tokens.

We parameterize $f_\phi(\cdot)$ with an LLM. While the pre-trained LLM can easily handle all the text tokens $\mathbf{a_{1:i-1}}$, it must be adapted to handle vision tokens corresponding to frames $\mathbf{v_i}$. We construct a video projection layer to process the video component and project it into the LLM's representation space. The overall causal problem formulation of contextual video captioning becomes

$$\hat{\mathbf{a}}_\mathbf{j}^\mathbf{i} = f_\phi \left( g_\theta(\mathbf{I}(v_j^1), \ldots, \mathbf{I}(v_j^k)), \mathbf{a}_\mathbf{j}^{\mathbf{1:i-1}} \right) \qquad (1)$$

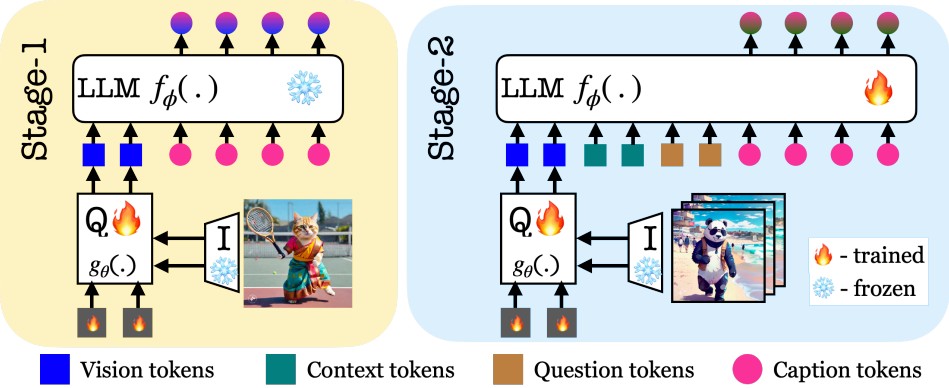

Figure 2: **CALVIN**: The architecture has 3 main components. (1) A frozen image embedding extractor I, (2) Non-linear projection module Q, and (3) An LLM. We train the model in 2 stages. Stage 1, we train only the projection module Q on image caption data. Stage 2, we use instruction formatted higher quality image-video data and finetune the parameters of Q and LLM. Refer to Sec. 3 for more details. Image and video examples shown here are synthetically generated using Meta Imagine [1].

---

**Data format for training:**
**Stage 1:** `<visual><caption>`
**Stage 2:** `<visual>` `<context>` `<question>` `<caption>`

**Examples:**
**Stage 1:** `[IMAGE_TOKENS]` `A cat playing tennis in a saree.`
**Stage 2:** `[VIDEO_TOKENS]` `Leonard the panda has a day off.` `He went for a swim.`
`How would you describe the key visual elements and actions in this video?`
`Leonard is taking a stroll down the beach`

---

Figure 3: **Stage-wise data format example**. We present the templates and an example of data format for different stages as shown in Figure 2. The colors of the text in this example match the token type colors in Fig. 2. The underlined text represents the segments that the language model must predict at all stages, and it is upon these predictions that the loss is computed.

where $\mathbf{I}$ is a *frozen* image embedding model that accepts an image frame $v_i^k$ and outputs a fixed $d-$dimensional embedding. The function $g_\theta(\cdot)$ is a *learned* non-linear projection module that takes in multiple image embeddings and projects them into the LLM latent space. We use a CLIP ViT-h/14 [99] vision encoder as the frozen feature extractor $\mathbf{I}$.

We train the system to predict the next text token based on all available context in the scene:

$$\theta^*, \phi^* = \arg\min_{\theta,\phi} \sum_{j=1}^{n} \sum_{i=1}^{m} \mathcal{L}\left(\hat{\mathbf{a}}_{\mathbf{j}}^{\mathbf{i}}, a_j^i\right)$$

where $\hat{\mathbf{a}}_{\mathbf{i}}$ is as defined in Equation (1) and $\mathcal{L}$ is cross-entropy loss.

Our projection module $g_\theta(\cdot)$ is comprised of two sub-components: a Q-Former layer [47], and a linear layer. The Q-Former begins with a fixed number of learnable embeddings, which are cross-attended by the video-frame embeddings during training. This formulation provides flexibility in handling a variable number of CLIP embeddings, up to a maximum defined by the position embedding, hence allowing training with mixed image and video datasets. The Q-Former layer is initialized with the pre-trained weights of BERT base [17]. We add position embeddings to the video-frame embeddings before the cross-attention to capture the temporal dynamics in videos. The second part of the projection module is a linear layer that projects the output of the Q-Former into the latent space of the LLM. We use a Llama-2 7b variant [85] model as our base LLM module.

## 3.2 Data and Stage-wise Training

The MAD dataset [80] stands out as the only extensive audio-description dataset currently available that includes aligned visual data, making it uniquely suitable for training contextual captioning

| GT: valjean strokes cosette's hair | GT: the moon shines through trees in the forest | GT: marius exchanges a glance with enjolras and follows her | GT: cosette sees a scar on his wrist |
|---|---|---|---|
| CALVIN: in a carriage, victor sits with his wife's head resting on his lap as he gently strokes her hair | CALVIN: a full moon shines in the night sky as victor walks through a dense forest | CALVIN: in the tavern, william glances at a woman as he walks past her | CALVIN: in a close-up, the man's hand grips the woman's wrist |
| CALVIN-3S: valjean strokes her hair | CALVIN-3S: valjean and cosette ride through the night in a carriage | CALVIN-3S: marius follows her | CALVIN-3S: valjean takes cosette's hand |

Figure 4: Qualitative results: Captions based on previous context are denoted **CALVIN-3S**. Captions without any context are represented as **CALVIN**. Contextual captions are close to ground truth and can get the names right based on the context. Without context, we see hallucinations (underlined), especially with the names.

models. However, the dataset's scale is relatively modest, especially when considering the extensive data requirements for effectively training a video-based LLM. To address this limitation, we train jointly on both image and video data. We use a single unified CLIP and Q-former vision pipeline for both modalities, enabling us to seamlessly transfer knowledge from the image to the video domain. We use a two-stage training process for the video LLM, as illustrated in Fig. 2.

In the first stage, we align the projection component (as denoted by **Q** in Figure 2), which consists of the Q-Former and a linear layer, while keeping both the CLIP and LLM frozen. In this stage, we use an internal dataset of image-caption pairs ($\approx$ 100M pairs), CC-12M [76] and the pretraining stage data curated by LLaVA [58, 59] which is built on top of MS-COCO [56].

For the second stage of training, we fine-tune the projection component as well as the LLM as shown in Fig. 2. This stage also marks our transition to employing higher-quality vision datasets. The data mixture for this phase is significantly enriched, comprising CC-3M [76], instruction tuning data from LLaVA 1.5 [58] – itself a curation based on a range of public image-caption or QA datasets including OKVQA [62], A-OKVQA [74], TextCaps [79], Visual Genome [40]. Additionally, we incorporate the WebVid-3M [8] video caption dataset and MAD [80] dataset train split into this stage. As such, this data mix is of higher quality as the majority of it is human-annotated.

Contrary to the approach of AutoAD [30] where the MAD dataset was trained separately, our ablation studies, which will be discussed later, reveal that integrating the MAD dataset into the second stage speeds up and simplifies training without significantly impacting the performance of the final model.

**Turning Stage-2 data into instruction data:** The dataset from LLaVA-1.5 comes pre-formatted for instruction tuning, simplifying its integration into our training process. For the other datasets, we adapt them to align with this instruction format. Typically, these datasets follow a `<image/video><caption>` structure. We restructure them into a more comprehensive format: `<image/video><context><question><caption>`. In this format, for all datasets except MAD, the `<context>` component is simply a placeholder space. For MAD, we use the annotation from the previous few scenes as `<context>`. We sample from a curated set of template questions regarding the `<question>` component. For a detailed view of these template questions, please refer to the Appendix.

## 4 Experiments & Results

**Training details:** All models are trained on a single A100 node with 8 GPUs. In Stage 1, we train only the projection module (Q-Former and linear layer) for $400,000$ iterations, with gradient accumulation over 4 steps and per-GPU batch size of 32. The learning rate has a cosine schedule and a warmup phase of 2,500 steps with min LR $1e-6$ and max LR $1e-4$. In Stage-2 we train Q-Former, linear projection, and the LLM. We train each model for $120,000$ iterations with a cosine learning rate with min LR of $1e-6$ and max LR of $1e-4$. The per-GPU batch size is 12 for image datasets and 6 for video datasets. Across all stages, a weight decay of 0.05 was applied. We adopt the Low-Rank Adaptation (LoRA) [34] approach for training the LLM. We set the LoRA rank to 32 and use LoRA on the QKVO (Query, Key, Value, and Output) in the attention layers. Unless stated, the Q-Former contains 4 layers with 32 learned embeddings and is trained to accept a maximum of 32 frames from the video. During the training, we sample 32 frames from the video. The datasets used in each stage are discussed in detail in Sec. 3.

**Metrics:** We evaluate our models and compare against baselines on four metrics. The first is BertScore [109], which measures the similarity in BERT representations between the ground truth and the generated captions. The remaining three are traditional captioning evaluation metrics, CIDEr [86], ROUGE-L [53] and, SPICE [6] each offering a unique evaluation perspective.

**Evaluation details:** We benchmark against the current state-of-the-art model, AutoAD-II [31], using the MAD-named-test split. This test split has captions that include character names. Owing to

the unavailability of trained models for both Auto-AD [30] and AutoAD-II [31], we present the best-reported results from these papers. Other baselines include ClipCap [65] and CapDec [67] which were discussed in Auto-AD I.

A limitation of the MAD dataset [80] is the absence of raw videos; only embeddings of the CLS token of the ViT model are shared. This constraint prevents the evaluation of models that require data in formats other than these specific embeddings. To provide a broader assessment of our model's performance relative to general-purpose open-source video language models, such as MovieChat [81], VideoLLaVA [52] and VideoLlama [108], we have conduct zero-shot comparisons on the TV-captioning (TVC) dataset [43].

## 4.1 Results

**MAD-eval.** Our main objective is to do contextual captioning. Hence we compare CALVIN against the SOTA models on the MAD-eval dataset in Tab. 1. Across all the metrics, we see CALVIN has significantly higher performance compared to the SOTA model. Even though our model is trained with context, we present a baseline case where we evaluate without context. In this case, we see ~90% improvement over the best baseline BertScore. Among models with context, we see a further ~26% improvement on CIDEr and ~68% improvement on BertScore. We present a few qualitative results in Figure 4. CALVIN-3S represents the case with 3-scene context and CALVIN is the evaluation with no context. When evaluated with no context, the model hallucinates names in some scenarios (scenes- a, c). However, these hallucinations disappear once we provide context to the model and the captions are quite close to the ground truth. For scene b, our model's generation differs from the ground-truth caption but is technically correct. We encounter a few such false negative cases where a human would find this generation acceptable but it is hard to identify such cases without human intervention. We present some such scenarios in the Appendix.

**TV-Captioning.** We also benchmark CALVIN against off-the shelf SOTA video-LLMs on the TVC captioning [43] task in Table 2. All models are evaluated in zero-shot fashion, where none of the models (including ours) are fine-tuned on this dataset. Context is not available in this dataset, but this dataset's captions refer to the characters in the scene with names, and so we add the character names in the prompt when we are querying all the models. CALVIN outperforms all the baselines across all metrics despite not being trained on this dataset. The distinction is quite visible on the CIDER metric where CALVIN is ~3x better than the next best model. This shows the generality of our model for movie/TV captioning tasks. We discuss a few ways of contextualizing captioning when the context data is not available in Section 5.

Table 1: **Evaluation on MAD-named-eval split.** The top half represents models evaluated without context. The bottom half shows the models trained/evaluated with context. Context column - the numbers in brackets show the number of scenes used in context. †- The numbers are from the original papers as the models are not public.

| Model | Context | BertScore ↑ | CIDER ↑ | ROUGE-L ↑ | SPICE ↑ |
|---|---|---|---|---|---|
| ClipCap [65] | No | 11.8 | 4.4 | 8.5 | 1.1 |
| CapDec [67] | No | 14.3 | 6.7 | 8.2 | 1.4 |
| CALVIN (Ours) | No | **27.25** | **14.74** | **11.9** | **3.89** |
| AutoAD-I [30]† | Yes (6S) | 23.8 | 21.9 | 13.9 | 4.8 |
| AutoAD-II [31]† | Yes (Char.) | - | 19.5 | 13.4 | - |
| CALVIN (Ours) | Yes (3S) | 39.08 | 25.47 | 16.30 | 7.33 |
| CALVIN (Ours) | Yes (5S) | **40.18** | **27.71** | **16.83** | **7.76** |

Table 2: **Zero-Shot evaluation on TV-Caption dataset.** All the models are provided with the names of characters in the scene. All the models use 7B LLMs.

| Model | Zero-Shot | BertScore ↑ | CIDER ↑ | ROUGE-L ↑ | SPICE ↑ |
|---|---|---|---|---|---|
| VideoLlama [108] | Yes | 28.29 | 3.34 | 5.61 | 3.52 |
| MovieChat [81] | Yes | 38.11 | 8.43 | 12.09 | 9.21 |
| VideoLLaVA [52] | Yes | 48.44 | 12.4 | 15.8 | 10.9 |
| CALVIN (Ours) | Yes | **52.16** | **38.9** | **20.1** | **14.27** |

Table 3: **Model Ablations:** Unless otherwise stated, all models are trained for the same number of iterations and on the same dataset. Unless otherwise stated, all the models are evaluated with a 3-scene context on MAD-named-eval split. ★-refers to the CALVIN 7B variant which we discuss throughout the paper

| Ablation type | Config. | BertScore ↑ | CIDER ↑ | ROUGE-L ↑ | SPICE ↑ |
|---|---|---|---|---|---|
| Stage-2 Data | All ★ | 39.08 | 25.47 | 16.30 | 7.33 |
| | (-) MAD | 17.94 | 3.52 | 7.26 | 1.44 |
| | (-) WebVideo | 37.35 | 22.07 | 15.5 | 6.7 |
| | (-) Image VQA data | 34.99 | 20.21 | 14.53 | 6.3 |
| | (-) All stage-2 data | 16.20 | 2.24 | 7.14 | 1.24 |
| Component finetuning | Q-Former | 33.59 | 15.91 | 14.22 | 5.28 |
| | LLM | 11.18 | 0.1 | 0.4 | 0.3 |
| | Q-Former + LLM ★ | 39.08 | 25.47 | 16.30 | 7.33 |
| Q-Former tokens | 32 tokens, width=4 ★ | 39.08 | 25.47 | 16.30 | 7.33 |
| | 64 tokens, width=4 | 38.83 | 24.27 | 16.22 | 6.94 |
| | 128 tokens, width=4 | 38.97 | 23.10 | 16.35 | 6.97 |
| LLM tuning | LLM + prefix tuning | 28.99 | 11.46 | 11.89 | 3.87 |
| | LLM + Lora (rank=16) | 38.63 | 24.67 | 16.08 | 7.09 |
| | LLM + Lora (rank=32) ★ | 39.08 | 25.47 | 16.30 | 7.33 |
| | LLM + Lora (rank=64) | 39.27 | 24.73 | 16.57 | 7.18 |
| Stagewise | MAD in Stage-2 ★ | 39.08 | 25.47 | 16.30 | 7.33 |
| | MAD in Stage-3 | 39.34 | 23.46 | 16.34 | 7.05 |
| Stage-2 Learning Rate | LR=$1e-5$ ★ | 39.08 | 25.47 | 16.30 | 7.33 |
| | LR = $1e-4$ | 34.59 | 15.91 | 14.22 | 5.28 |

## 4.2 Ablations

We share a few ablations and some insights gained. Since there are many moving parts in the system, the search for our final configuration is mostly greedy and looks at one component at a time. We present the captioning metrics of the resulting models in Table 3. All models in the table are trained for the same number of iterations with a context of three previous scenes along with the video. These models are also evaluated with a 3-scene context, and other hyperparameters are kept constant.

**Data mixture in Stage-2.** As discussed in Section 3.2, we combine multiple datasets in this stage. We remove one data type at a time. Removing Stage-2 training impacts the performance the most, followed by removing the MAD dataset. This makes sense since AD captions tend to be crisp and more contextual, while others are more descriptive. There is a clear domain shift and we need the MAD dataset in training to bridge this gap. One interesting observation is that removing the image VQA data impacted the performance a bit more than removing the only video dataset other than MAD. This confirms that the image VQA data can contribute to video understanding.

**Component tuning.** As discussed in Section 3.2, we tune both Q-Former and LLM in Stage-2. We check performance when just one of these components is present. It turns out that training only the LLM leads to an extreme drop in performance while training only Q-Former dropped performance slightly, especially the CIDER score. We believe in LLM training case, the model completely ignores the vision embedding, attending only to the context, causing generated captions that are hallucinations with no grounding in the video. In the case of only Q-Former training, we believe that the model does not learn to utilize the context well, hence a slight degradation in performance.

**Q-Former tokens.** We examine the effect of the number of Q-Former tokens, which in turn controls the parameter count. More tokens causes a slight degradation in performance. We believe this is because the smaller size of the stage-2 dataset causes overfitting.

**LLM tuning parameters.** We examined two types of efficient LLM fine-tuning techniques - LoRA and Prefix Tuning [50]. Prefix-Tuning adds a few additional trainable parameters to each transformer block. In LoRA, a chosen set of parameters is updated by a low-rank approximation. First, we see LoRA training is significantly better than Prefix-Tuning. Second, as we increase the LoRA rank hyperparameter, we see slight performance improvement.

**Other ablations.** We depart from previous works by mixing the MAD dataset into stage II rather than separately fine-tuning on it. If we instead fine-tune on MAD with a 40,000 iteration stage III, we do not see much improvement. Additionally, we observe that reducing the max learning rate from $1e-4$ to $1-5$ in Stage 2 improves the performance by a non-trivial amount. We refer the reader to the Appendix for additional train-time and inference time ablations.

> **Prompt formats:**
> No context: `[VISION_TOKENS]`. Describe this visual.
> Entity context: `[VISION_TOKENS]` The characters and entities in the scene are `<ENT1>` and `<ENT2>`. Describe this visual.
>
> **Example model outputs:**
> No context: She is talking to a blonde man.
> Entity context: `<ENT1>` is talking to `<ENT2>`.

Figure 5: **Entity Prompting.** Instead of the previous scene context, we prompt the models with only the entity information. Refer to Section 5.1 for more details.

# 5  Video-language models are also few-shot learners!

It was shown in Brown et.al [10] that LLMs, trained on a diverse set of data, can benefit from prompt engineering and in-context learning. Despite our video representations being trained in the specific domain of video captioning, they still inherit the emergent properties of the parent LLM. This includes their ability to adapt to prompt engineering, in-context learning, and fine-tuning for specific tasks. In this section, we propose and evaluate two realistic strategies for customizing the model at test time. These strategies have shown improved performance in captioning compared to scenarios where no context is provided.

> *Regularization data*: I don't recognize the characters or entities in the scene. Hence the final caption is: `<MODIFIED_ORIGINAL_CAPTION>`.
>
> *Test-time movie data*: The characters and entities in the scene are `<ENT1>` and `<ENT2>`. Hence the final caption is: `<ORIGINAL_CAPTION>`.

Figure 6: **Few-shot fine-tuning**. We rewrite the captions in CoT [92] style and fine-tune the model on them.

## 5.1  Prompt engineering

It is hard for the models (and humans) to associate actors with their characters' names without a priori information. We observe that our model can be enhanced further by providing characters' or entities' information in the context. Assuming we can access entity information for a given scene, we prompt the model as illustrated in Figure 5. We noticed that just adding this information about entities in the scene to the context improves performance. See Section 5.3 for results.

## 5.2  Fewshot finetuning

Works like Dreambooth [72] "personalize" a large pre-trained model to an object or a category with test-time training. We test this idea by fine-tuning our model for every movie in the test set, showing it 20 scenes (∼5 minutes) of each movie. This paradigm enables human-in-the-loop annotation where the humans can correct the model's mistakes while the model performs the bulk of the work.

To avoid over-fitting on the test time data, we propose to use additional regularization data from the train set (*i.e.* pre-annotated other movies) where we strip away the entity details using LLaMA-2 70B to get a rephrased caption without proper nouns or identifying details. See Figure 6 for an example of test-time movie data and regularization data used for test-time training. This is analogous to *chain-of-thought* prompting [92], where a model is trained to output its reasoning. In our use case, we show that with fine-tuning, the model recognizes some of the main characters as well as adapting to the tone of annotation for the movie. Note that annotators use different styles or languages based on the type of the movie. For instance, animated films have simplistic audio descriptions as they are directed at children. We share results in Section 5.3.

**Experiments.** We first sample 20 scenes from the test-time movie such that the scenes in which main characters first appear are included (see Table 4 for 50 instead of 20). Then we add the same number of samples for the regularization data, which are sampled randomly from the MAD-train split. We rephrase the captions of the whole mix in *chain-of-thought* (CoT) style as present in Figure 6. We start with CALVIN Stage-2 checkpoint and finetune it for 100 iterations with a constant learning rate of $1e-5$, 2-step gradient accumulation, and batch size of 10 on a single A6000 GPU. Each model training run takes less than 6 minutes.

| | |
|---|---|
| **GT:** Lisa takes a few steps toward him and smiles. | **GT:** shaking his head, george shrugs his hands and smiles. |
| **CALVIN:** She smiles at him and walks away. | **CALVIN:** a smiling man wearing a blue suit and a pink tie walks down a hallway in an office building. |
| **CALVIN-E:** She wears a tight-fitting black dress with a plunging neckline, and her blonde hair is styled in loose curls | **CALVIN-E:** George smiles and shakes his head as he walks away. |
| **CALVIN-FS:** Lisa smiles at him. | **CALVIN-FS:** George smiles and nods. |

Figure 7: **Qualitative results:** We present two adjacent scenes from one of the test movies. **CALVIN** corresponds to the result with no context. **CALVIN-E** refers to context containing only entity information. **CALVIN-FS** corresponds to the model few-shot trained on 20 random scenes from this movie (not the ones shown here.)

## 5.3   Results

We present the average metrics across all 10 movies in Tab. 4. We present results on CALVIN without context and CALVIN with entities in the prompt as discussed in Sec. 5.1. We present 2 fine-tuning scenarios trained with 20 to 50 samples from each test movie. Note that this accounts for less than 8% of total scenes with AD in all of the test movies.

We see performance improvement in both the personalization scenarios over the model without context. This shows that test time personalization can be a cheap and efficient way to improve contextual captioning. We present a qualitative example for 2 continuous scenes in Fig.7. The model without context often describes scenes in a generic albeit correct way. CALVIN with entities in the context is able to caption the scene a bit better, however, the model does not always use this information, e.g., the left scene caption does not refer to the character 'Lisa'. CALVIN-FS (fewshot) captions well with correct character names. See Appendix for ablations on the amount of training and number of few-shot examples. As a downside, few-shot finetuning seems to reduce caption diversity and length and occasionally associates wrong names with characters.

Table 4: **Test-time adaptation results:** (first row) CALVIN evaluation without context. (second row) 'Entities' means the context has just the list of entities in the scene as discussed in Sec. 5.1. (third and fourth rows) For few-shot training from Sec. 5.2, the number in brackets counts examples used in finetuning. Both strategies improve performance over the no-context.($^{\dagger}$ Numbers differ slightly from Tab. 1 since the numbers presented here are an average of metrics computed one movie at a time, and some metrics like CIDEr depend on the word distribution of evaluation set.)

| Model variant | Context | BertScore ↑ | CIDER ↑ | ROUGE-L ↑ | SPICE ↑ |
|---|---|---|---|---|---|
| CALVIN $^{\dagger}$ | None | 26.91 | 10.85 | 11.76 | 3.73 |
| *Prompting §5.1* | | | | | |
| CALVIN | Entities | 39.14 (+12.23) | 16.27 (+5.42) | 15.56 (+3.8) | 7.13 (+3.4) |
| *Few-shot training §5.2* | | | | | |
| CALVIN + FS (20) | None | 34.14 (+7.23) | 17.74 (+6.89) | 14.06 (+2.3) | 7.17 (+3.44) |
| CALVIN + FS (50) | None | 36.19 (+9.28) | 20.06 (+9.21) | 15.22 (+3.46) | 7.21 (+3.48) |

## 6   Discussion and Conclusion

We present a state-of-the-art model for contextual scene captioning. Our model introduces several innovations, including instruction tuning and stage-wise training for various components, along with tailored data mixes at each stage. We demonstrate that our model, CALVIN, exhibits superior generalization capabilities and outperforms existing off-the-shelf video-LLMs at zero-shot evaluation on a TV-captioning dataset. We additionally propose novel test-time adaptation strategies involving prompting and few-shot tuning.

There are still ample opportunities for enhancement. These include processing longer videos, intelligent frame sampling for better and faster video representations, and audio processing. The biggest challenge to these advancements is data scarcity; There is a pressing need for public video datasets of high quality.

## 7   Acknowledgements

This work was made possible by National Science Foundation (NSF) grant #2213335. Further support was provided by the National Science Foundation (IIS-2212182), and by the NSF TRAILS Institute (2229885).

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

# CALVIN: Improved Contextual Video Captioning via Instruction Tuning
## Supplementary Material

## A  Data curation additional details

All the human annotations in this paper are done exclusively by the authors. **Data Cleanup:** During the initial training of our models, we encountered an unexpected challenge. Despite using datasets generally regarded as high-quality, we observed that the models were outputting repetitive patterns of numerical data. A more detailed examination revealed that the captions in the WebVideo dataset often included specific dates or video resolutions, like 1930x1080 or 4HD. These elements were inadvertently leading the model to generate dates and resolutions in its output. We scrubbed numerical data from WebVideo using hand-crafted (regex) functions, resulting in a marked improvement in the quality of the model's output.

We also identified that the bounding box coordinate questions in the instruction fine-tuning dataset of LLaVA-1.5 [58], which are uncharacteristic of our problem domain, proved detrimental to the downstream captioning task. Hence, we excluded these.

Lastly, we noticed that MAD includes movie credits at the beginning or end of some films. We used LLaMA-70B [85] with a few human-annotated in-context examples to tag training examples as either credit-related or not. Removing scenes with credits caused a subtle improvement in the quality and relevance of the generated captions.

## B  Stage-2 training additional details

In Tab. 5, we present the question templates used in training of CALVIN Stage-2 model, to convert video-caption data into instruction data.

Table 5: Stage-2 question templates

| Question template |
| --- |
| What is this video about? |
| Describe the video, including the actions and scenes. |
| Provide a description of the given video, capturing its key moments. |
| Give a concise explanation of the video you see, including the events and characters. |
| Summarize the contents of the video, focusing on the main events and participants. |
| Detail the scenes and characters present in the provided video clip. |
| Narrate the sequence of events in the video, including significant actions. |
| Report on the events and individuals portrayed in the video clip. |
| Highlight the pivotal scenes and actions observed in the video. |
| What's going on in the video? |
| Describe the scene in the video |
| Summarize the key visuals and events of this video. What's happening in the video? |
| Detail the primary actions and visuals of this footage. |
| Provide a brief account of the scenes and characters within this video. |
| Elucidate the main events and visuals in this clip. |
| Give an overview of the storyline and characters present in this video. |
| Narrate the visual elements and the main events showcased in this video. |
| Chronicle the character dynamics and scene transitions in this video. |
| Describe the aesthetic elements and main occurrences in this footage. |
| Highlight the significant actions and interactions within this video. |
| Offer a concise breakdown of the scenes and events in this clip. |
| Detail the setting, characters, and major events of this video. |
| Explain the visual motifs and character roles within this video. |
| Decode the main story arc and visuals in this footage. |
| Provide an interpretation of the character relationships and visuals present. |

# C Qualitative analysis of CALVIN vs ground truth

In Tab. 6, we present a few examples of predicted caption vs ground truth caption, one from each movie in MAD-eval split. Clearly CALVIN-3S captions are as good as GT caps in many scenarios.

In the main paper Fig. 4(b), we have seen that the GT caption and CALVIN-3S captions differ quite a bit, however, the caption is an acceptable alternate when we looked at the video. We present a few cases from one of the MAD-eval movies, HOW DO YOU KNOW? in Tab. 7. This shows that the maximum achievable bert-score on this task is lower than 100 due to the subjective nature of this task.

Table 6: **Ground truth vs Predicted captions on CALVIN-3S model:** We present an example from each movie in the MAD-eval dataset.

| Movie | Ground Truth Caption | Predicted Caption |
|---|---|---|
| IDES OF MARCH | stephen closes the door behind him | stephen shuts the door behind him |
| HOW DO YOU KNOW? | lisa presses the elevator button | lisa presses a button on the elevator's control panel |
| THE ROOMMATE | sarah smiles as stephen leans in and kisses her | sarah leans in and kisses stephen |
| BATTLE LOS ANGELES | joe nods at nantz | joe nods, then turns to nantz |
| Harry Potter and the goblet of fire | dumbledore thrusts it into harry hand | dumbledore hands it to harry |
| CHARLIE ST CLOUD | charlie eyes pool with tears | charlie's eyes well up with tears |
| LEGION | gabriel punches michael to the floor | michael punches gabriel in the face |
| LES MISERABLES | marius gently kisses eponine brow | marius and eponine kiss passionately |
| HANSEL GRETEL WITCH HUNTERS | hansel falls to the ground | hansel climbs down from the tree and collapses on the ground |
| SIGNS | graham stares blearily down at the floor | graham's gaze drops to the floor |

Table 7: **Human acceptable predictions:** Some ground truth captions vs CALVIN-3S generated captions which the authors felt are acceptable despite being different from the GT. All the examples are from the MAD-eval movie, **How do you know?**

| Ground Truth Caption | Predicted Caption |
|---|---|
| the elevator door shut | matty watches her go |
| george accepts the letter | she hands him the letter |
| lisa nods | lisa's mouth hangs open as she stares at him |
| in george office | the bespectacled man shakes his head |
| in his suit and dress shoes | george sprints down the sidewalk |
| annie loads the fridge | she sets the bag on the kitchen counter |
| now on the phone | george hangs up the phone |

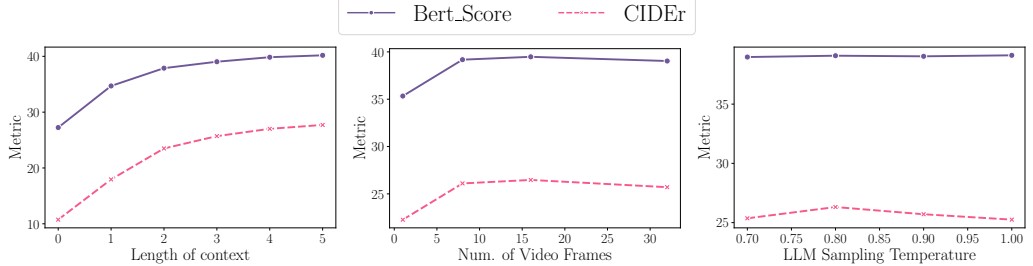

Figure 8: **Inference time ablations.** Effect of inference-time hyperparameters. (Left) Number of previous scenes in the context (Middle) Number of frames sampled per scene (Right) LLM sampling temperature.

## D    Additional ablations.

**Inference-time ablations.** In Figure 8, we show how BertScore and Cider metrics change with context length, number of video frames sampled, and LLM sampling temperature. We see that increasing the number of previous scenes in the context improves performance across both the metrics; The model was trained with just 3 scenes in the context and yet generalizes to more. Regarding the number of frames sampled from each clip, we see performance improvement as we go from 1 frame to more, indicating that the model uses motion information in caption generation. However, we see no improvement in performance beyond 8 frames. LLM sampling uses beam search. We do not see much difference in BertScore at lower temperature, but we see a slight bump in CIDEr.

## E    Fewshot-training ablations

We conducted an ablation on one movie from the MAD-train split - `3041 - JUST GO WITH IT` to understand the right number of samples and training iterations needed for few-shot experiments. We ablate the number of iterations in Fig. 9 and the number of annotated samples in Fig. 10. We observe that the performance does not improve beyond 50 annotated samples in training. And we see that training for more than 100 iterations does not improve the performance.

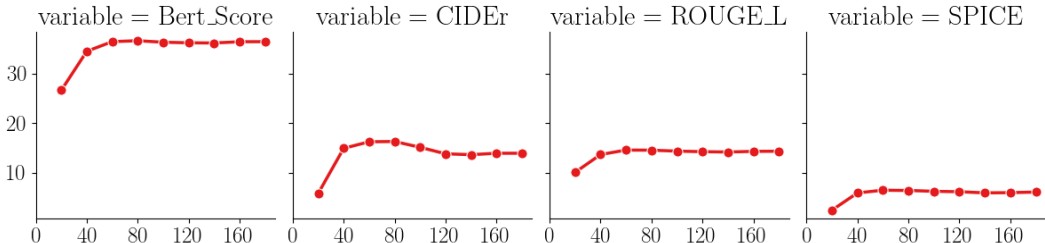

Figure 9: Number of training iterations vs Metrics.

## F    Broader Impacts.

**Broader Impacts** By leveraging previous scene contexts, CALVIN is well-positioned to enhance the accessibility of visual media for individuals with visual impairments, offering them a more immersive and contextually rich experience. This improvement could make entertainment and educational content more inclusive, promoting equal access to information and enjoyment regardless of visual capability. Furthermore, the technology could be applied in various other domains such as automated content moderation, where understanding the context of scenes can improve the detection of inappropriate content, and in digital humanities, where researchers can analyze films at scale to study cultural representations and evolution.

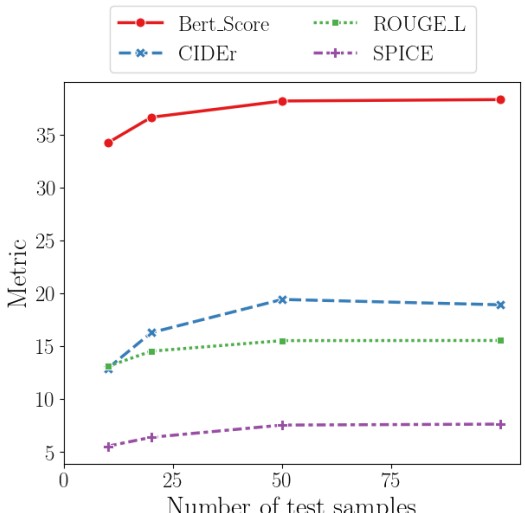

Figure 10: Number of ground-truth samples vs Metrics.

