# OpenReview forum: "CALVIN: Improved Contextual Video Captioning via Instruction Tuning"
_NeurIPS.cc/2024/Conference — NeurIPS 2024 poster_

### Official Review · Reviewer_GsDL · 2024-06-30

**Soundness:** 3
**Presentation:** 3
**Contribution:** 2
**Rating:** 5
**Confidence:** 3

**Summary:**

The paper focuses on captioning movie scenes, which unlike typical captioning tasks, videos from movies can be captioned in a way that tells the story of the movie. For example, captions from movie scenes should convey how the characters felt, and not just the detail in the image.  To be able to caption in this way, the authors propose and train CALVIN, a 7B LVLM, on MAD (a movie-audio dataset) as well as other datasets. They then experiment with a variety of techniques to experiment with CALVIN in various settings.

**Strengths:**

* The core contribution of this paper is a large model to caption movie scenes, which is meaningfully different from captioning typical videos. The weights of the model can contribute to further study this difference, or further explore movie captioning.
* Big improvements over existing work
* There are lots of experiments and ablations, testing the model in prompting, test-time setups, and even few-shot finetuning (i.e., personalization toward a specific movie)

**Weaknesses:**

* I am not sure I can say there’s technical novelty in this work (this is not a deal breaker, since existing knowledge was applied to create a research artifact that did not exist prior). However, I do not see a statement about releasing weights (only code), and if it will not be released, I am not sure what the contribution of the paper is.

**Questions:**

What do you do with frames that introduce new information specific to the movie? For example, how could a model correctly predict the ground truth for frames that introduce names of characters?

**Limitations:**

I do not see where in the paper core limitations of CALVIN or the methodology are discussed.

---

> ### Author Rebuttal · Authors · 2024-08-07
>
> Thank you for your detailed feedback, GsDL. We're happy to answer your questions regarding our paper's key contributions, model weight release, and how to adapt the model to new movies.
>
> **[Main Contribution of the paper]** While the model weights are important (and we are working to release them), our work also includes a number of key contributions concerning architectural choices for video LLM, model training, and dataset choice and setup. On the architectural side, we show that we can simplify/unify complicated input pipelines from prior work with a single Q-former, and we highlight the importance of continued training in the LLM part of the model.
>
> We believe this can help video-understanding researchers make calls while training the next-gen models. For example, we are one of the few video-understanding papers that calls out the importance of data cleaning and data mixtures which is also echoed by recent works such as MM-1 [1], and PaliGemma [2]. We further introduce novel few-shot adaptations of the model for new movies which can make the model useful in practice and take it beyond the academic realm.
>
> **[Weights release]** This issue is currently under legal review. As a backup option, we are also currently working with a partner to release a non-official version of the weights for the camera-ready.
>
> **[How to predict new information specific to the movie?]** Great question. This is exactly what we tried to answer in Section 5. In our experiments, we found CALVIN was predicting character emotions and physical states well but it is hallucinating the names and locations when we do not provide the context on a new movie. Hence in section 5.2, we few-shot finetune on some scenes with main characters and ground-truth ADs, this led to non-trivial improvement in the performance even without context. We believe this could be a way to adapt CALVIN for new movie audio descriptions.
>
> [1] - McKinzie, Brandon, et al. "Mm1: Methods, analysis & insights from multimodal llm pre-training." arXiv preprint arXiv:2403.09611 (2024).
>
> [2] - Beyer, Lucas, et al. "PaliGemma: A versatile 3B VLM for transfer." arXiv preprint arXiv:2407.07726 (2024).
>
> -------
> Thank you again for your thoughtful review. We hope we addressed your concerns/questions sufficiently and we would appreciate it if you would consider raising your score in light of our response. Please let us know if you have additional questions we can address.

---

> > ### Comment · Reviewer_GsDL · 2024-08-12
> > **Thank you**
> >
> > Thank you for the response.
> >
> > **Main Contribution of the paper** Thank you for clarifying. I do maintain my belief however, that the implementation of CALVIN is not novel. Careful data cleaning (and data mixtures) is not a specific issue for video and there are countless papers that study its importance in adjacent fields, such as LLM pretraining. To the best of my understanding, the core contribution of this paper is not to study the importance of quality data in training, nor does it contribute tools to clean data specific to video, so I do not see how this furthers the field of video understanding on the axis of data cleaning. Similarly, if the main point of the paper is to show a single Q-former is enough to replace a complex pipeline, then I do not believe the text reflects that. For example, the abstract does not indicate any type of novelty on the training method, but it does emphasize the utility of CALVIN.
> >
> > Regarding the few-shot adaptations, I agree they are useful, but this is not data cleaning or related to the pretraining of the model (including the dreambooth-like setup).
> >
> > **Weights Release** I am glad to hear that, I do believe it has a non-trivial impact on the value of this work.
> >
> > **How to predict new information specific to the movie?** Thanks, this does answer my question.
> >
> > For the reasons above, I choose to maintain my score.

---

> > > ### Author Response · Authors · 2024-08-13
> > > **Thank you Reviewer GsDL**
> > >
> > > We thank the reviewer for your thoughtful feedback. We note your concerns and we will update the abstract and introduction to highlight a bit more on architectural and training innovations (LLM continued training and Fewshot fine-tuning) we introduce in the paper.
> > >
> > > Please let us know if you have any additional questions we can address.

---

### Official Review · Reviewer_Y2oh · 2024-07-12

**Soundness:** 3
**Presentation:** 3
**Contribution:** 3
**Rating:** 6
**Confidence:** 4

**Summary:**

This paper introduces a specialized video LLM named CALVIN, which leverages previous movie context to generate contextual scene descriptions. The authors achieve this by training their model on both image QA tasks and video captioning tasks within a unified framework. Experiments demonstrate that with mixed training data and context during inference, CALVIN outperforms the previous state-of-the-art in the audio description task.

**Strengths:**

1.	The paper is well-written and easy to follow.
2.	This paper explores the use of video LLM in audio description, demonstrating significant performance gains over the previous SOTA.
3.	The paper presents detailed ablation studies, which are helpful in understanding the effectiveness of CALVIN.

**Weaknesses:**

1.	The method appears similar to VideoLLaMA except for differences in data usage and the base LLM. The authors should provide a more thorough discussion on how their approach differs from previous methods.
2.	The experimental results in Table 1 and Table 2 are not entirely fair due to the use of different training data and pre-trained models. The data seems critical for CALVIN's performance, as shown in Table 3. Therefore, it would be better to align data usage with the compared methods or at least highlight the types of data used by other models.
3.	Is it practical to use context from captions, given that ground-truth captions do not exist? What is the performance of using generated captions as context for CALVIN?

**Questions:**

See weaknesses.

**Limitations:**

The authors have discussed the limitation of their work.

---

> ### Author Rebuttal · Authors · 2024-08-07
>
> Thank you for your detailed feedback, Y2oh, we're happy to answer your questions regarding a better description of the architectural differences between CALVIN and VideoLLaMA, and regarding further dataset ablations.
>
> **[Clarification of Differences to VideoLLaMA and other recent video architectures]** The first main difference between CALVIN and other models is how the video representations are projected onto the Language model space. While previous methods such as videoLLama used 2 projection Q-formers (with one component frozen and another trained from scratch) in sequence, we simplified the pipeline with a single Q-former that is trained from scratch. The second main difference is the LLM finetuning and the usage of LoRA. Our experiments showed that unfreezing the LLM in stage 2 can lead to significant performance gains, while previous methods such as VideoLlama had that component frozen throughout the training.
>
> Many recent works such as MM-1 [1], Cambrian-1[2], and PaliGemma [3] have shown that while some critical architectural choices such as positional embeddings or certain activations could be important, most of the gains came from the data mixtures. Compared to previous works, we are also the first ones to propose the use of image VQA data in training and the conversion of caption data into synthetic VQA via LLMs and to show that adding these into training can improve performance.
>
> Overall, thank you for bringing up this question. We will add this discussion to our related work section.
>
> **[Dataset comparisons]**
> We provide a short table on the types of data used in the last stage of training of important baselines in the following table.
>
> Model | Datasets used
> | -------- | ------- |
> VideoLlama | Webvid-2M, CC595k
> Auto-AD I |  CC3M, Webvid-2M, AudioVault-AD (3.3M), MAD
> CALVIN | CC3M, Webvid-2M, Llava (600k), MAD
>
> While the scale of data used in the training of all the models is almost the same, how the data is used varies across the models. We shared our data learnings in the paper (Appendix A), showing that cleaning the data or removing certain types of data can improve performance significantly. We also modified the data into instruction-tuning format and this led to gains.
>
> We also thoroughly ablated different types of training setups and shared our learnings in our paper. We believe while our architecture and data slightly differ from the baselines, they are of similar scale, and our smart data curation and our thorough studies led to the best freeze-train configuration to achieve the final model.
>
> With even more compute, we would have liked to run additional ablations of the other models with our data, but this is not something we could budget in. In any case, we've included the exact data mix for all models in Table 1 and Table 2 with an additional column, to clarify to readers that these models do differ in their pre-training data.
>
> **[How to do contextual captioning when ground truth context doesn’t exist?]** Excellent question. This is quite a plausible scenario and this is the inspiration for Section 5.  We saw a drop in performance when we used generated captions in context, however, the performance is still better than the baseline model as well as Calvin with no context. We present the contextual captioning with self-generated ground truth in the following table. (We only presented Cider since that is the only metric available in Auto-AD paper for this scenario.)
>
> Model | Context length| Cider
> | -------- | ------- | ------- |
> Auto-AD I | 3 | 14.5
> Auto-AD 2 | 3 | 19.5
> Calvin | 3 | 19.9
>
> We observed the main reason for this performance drop is that many ADs contain either the first names of the characters or locations and without this knowledge in context, it is hard for models to predict correct names, and they sometimes hallucinate. Please refer to Section 5 to see our proposals to get around this issue with few-shot finetuning.
>
> [1] - McKinzie, Brandon, et al. "Mm1: Methods, analysis & insights from multimodal llm pre-training." arXiv preprint arXiv:2403.09611 (2024).
>
> [2] - Tong, Shengbang, et al. "Cambrian-1: A fully open, vision-centric exploration of multimodal llms." arXiv preprint arXiv:2406.16860 (2024).
>
> [3] - Beyer, Lucas, et al. "PaliGemma: A versatile 3B VLM for transfer." arXiv preprint arXiv:2407.07726 (2024).
>
> -------
> We thank the reviewer for a thoughtful review. We hope we addressed your concerns/questions sufficiently and we would appreciate it if you would consider raising your score in light of our response. Please let us know if you have additional questions we can address.

---

> > ### Comment · Reviewer_Y2oh · 2024-08-13
> >
> > Thank you to the authors for their detailed response.
> >
> > Regarding the first concern, although the authors listed some differences, I still find the implementation of CALVIN not novel enough. However, I appreciate the efforts made in optimizing the method through data and structure tuning, which indeed contribute to better performance in practice.
> >
> > For the second question, I understand the challenges in making a fair comparison. However, including at least one such ablation study would strengthen the paper.
> >
> > Lastly, I am pleased to see the new results in this setting, as they will serve as a valuable baseline for future studies.
> >
> > Overall, I am inclined to raise my score to 6.

---

> ### Author Response · Authors · 2024-08-13
> **Thank you Reviewer Y2oh**
>
> We thank the reviewer for raising the score. We agree that the new experiment has further enhanced the analysis. We will include all the new results in the appendix of the camera-ready version.

---

### Official Review · Reviewer_suoE · 2024-07-12

**Soundness:** 3
**Presentation:** 3
**Contribution:** 3
**Rating:** 6
**Confidence:** 4

**Summary:**

This paper addresses the task of contextual video captioning, with a particular application emphasis on settings for film and tv, where audio descriptions can be useful for making these mediums more broadly accessible. Standard vision-language models are often verbose (problematic since audio descriptions must fit between dialogue) and are prone to hallucinations (e.g., of actor names, or of entities not present in the scene but may be thematically related). The authors propose CALVIN, a model that incorporates pretraining on a range of vision-language tasks (image QA, video captioning, etc.), instruction tuning to improve captioning behavior, and some final adaptations (e.g., prompt engineering, few-shot adaptation, etc.). They observe improvements on the MAD-eval (for film) and TVC (for television) datasets, and qualitative results show more terse descriptions with fewer hallucinations.

**Strengths:**

Overall, there are a number of strengths to this work:

`+` The domain of contextual description for films and television is important and can have broad positive impacts. It is also a relatively new space in the broader space of dense captioning.

`+` The proposed model examines some sensible explorations of the VLM design space for this task setting. Among other things, the adjustments to the training recipe (i.e., which data goes in which training stage) seem to have a good impact, in ablations.

`+` Benchmark results on two different datasets, important since MAD-eval only provides pre-computed features (due to copyright), so having an additional dataset without this restriction helps to better clarify differences with other SOTA VLM models.

`+` Shows good performance results, quantitatively and qualitatively.

**Weaknesses:**

However, there are some key areas of weakness, as follows:

`-` The paper focuses on reducing hallucinations and verbosity, but the metrics provided (across the model and model-free/n-gram based ones) do not necessarily correlate or show that this is the reason that the metrics have gone up. Similar prior work (e.g., AutoAD series) have done extensive explorations of metrics around character relevance, and stronger model-based metrics. Given that many of these metrics/analyses have been released by prior work, and the research focus of this paper, the empirical results would be significantly stronger if such analyses were performed for here (and ideally, in a consistent way to make an explicit comparison with prior work).

`-` The paper presents a large set of ideas, but the individual novelty of each element is not clear, and it would be good to have clearer comparisons with the closest prior work that introduced the relevant idea (in as much of an apples-to-apples fashion as is possible). As one example, the incorporation of IMDB metadata (for actors) as a small set of exemplars has been explored by prior work (specifically, AutoADII); an ablation based on this approach seems like it would be an important reference point for characterizing the relative improvement with the slightly different style proposed here.

`-` The overall task of audio description also cares about the localization of the captions along with the captions themselves. While prior work (e.g., in the AutoAD series) also considered this aspect, this is notably missing from this architecture. It is also a bit unclear why, given that time tokens have been explored for similar VLM models already. Relatedly, the paper would benefit from improving the clarity of the exact inputs, in terms of the temporal regions that are provided to the model for captioning and how these are selected. (There is some language around the choice of the few-shot examples, but this comment is more broadly speaking across all settings).

**Questions:**

Overall, this work is borderline, with a preliminary rating of borderline+. The discussion phase will be important for this work, so if the authors could address some of the areas for clarification that are raised above in weaknesses, particularly with respect to prior work in this space, this would be helpful towards finalizing the rating. Note that for many of the points the review above notes a specific example for illustration, but the comments do extend more generally.

**Limitations:**

The authors have mentioned some limitations of their method, but not in great detail (there is a short, general statement in the final paragraph of the conclusion). This can be expanded in the supplement, for example, addressing the limitation that the model does not seem to output a clear localization of the caption.

---

**Post-rebuttal update:** The authors largely addressed the concerns raised in the rebuttal phase. After consideration of this and other reviewers' comments, I've raised my rating further.

---

> ### Author Rebuttal · Authors · 2024-08-07
>
> Dear Reviewer suoE, thank you for your detailed review. We're glad that you found our contextual captioning model insightful. We do think that our thorough analysis of all parts of the video LLM pipeline, from improved data cleaning and handling, to clearly ablated training recipes and architectural simplifications are a key contribution of our paper taken together as a whole, and we believe that this paper would be a good resource for next-gen contextual video LLMs.
>
> **[Character analysis]** The character analysis performed in Auto-AD II is to check the performance of  their character recognition module not the character recognition in the final caption. We noticed that Bert score is highly correlated to getting the character name right in the audio description (as Bert embeddings seem to be biased towards proper nouns). Prompted by your question, we also performed an analysis on the percentage of times the model predicts the name correctly, and we saw a score of 72% for our best model. Please note that in the rest of the 28% of the time, in high proportion, the names are replaced with “he” or “she” which are not technically incorrect.  We are unable to compute this Auto-AD series of models since the model checkpoints nor the model outputs are available.
>
> **[Verbosity]**  Prompted by your question, we computed the token count using TikToken for Table 2 (since outputs of baselines from Table 1 are not available), we present the average token counts on the test set for the models in the table below
> Model | Avg num of tokens (lower better)
> | -------- | ------- |
> VideoLlama | 82
> MovieChat | 126
> VideoLlava | 64
> Ours  | 24
>
> **[Hallucinations]** What we noticed is, that most of the models suffer from “factual errors” as discussed in Liu et al [1]. To the best of our knowledge, there are no benchmarks or tools to evaluate hallucinations in a free-form generated caption on a random video. However, to address your question, we used Gemini Vision Pro as a judge to give a hallucination score between 1 to 10 for a given generation. We prompted Gemini with some examples from [1]. We took the video clip from Fig 1 of the main paper (a scene from the Battle Los Angeles movie), which in turn has 6 ADs associated with it. We present the average scores given by Gemini in the table below. We see Calvin is significantly better than other OS models in terms of the level of hallucination according to the Gemini model as judge rating.
>
> Model | Avg hallucination score by Gemini (lower better)
> | -------- | ------- |
> VideoLlama | 8.7
> MovieChat | 9.1
> VideoLlava | 6.8
> Ours  | 4.3
>
> We will extend all the above-mentioned analyses with full details to the appendix of the final version of the paper
>
> **[CALVIN comparisons to prior work]** Calvin and Auto-AD II share most of the training data and whatever differs, the scale of the data is the same. While AutoAD II has additional aspects of incorporating character and identity information, ours is a simplified VideoLLM that takes a scene representation and pure text context. We do think that it is a key strength of our work to look for general-purpose improvements only through contextual information, without specializing the system further with e.g. actor information. While using IMDB actor data in Calvin is an interesting thought, we would like to point out that, despite using that information, Auto-AD II’s results are worse than Calvin’s, highlighting the advantages of the generic approach. We believe our model’s high performance is due to data cleanup, conversion of existing data into instruction tuning data, and a well-tuned model due to our exhaustive ablations.
>
> **[Temporal localization of AD]** Thanks for raising this point. Both AutoAD II and Calvin take a truncated video clip from the movie and previous context to produce Audio Descriptions. We observed that whether an AD should be written for a given clip or not depends on three factors 1) the subjectivity of the annotator 2) the Length of the clip 3) How different the clip is from the previous scene.
>
> The AutoAD II paper also observed that pauses longer than 10 sec have an AD while pauses less than 4 sec most likely do not have an AD. We also observed similar trends. AutoAD II looks at 30-sec proposals along with audio and video data in the time interval and makes a prediction for each 5-seconds whether AD exists or not.
>
> We approached this problem slightly differently as we believed a simpler solution was possible. First, we only look at the “pauses” in the audio of the movie.  We trained a simple classifier head on top of our Q-former layer which classifies whether a clip needs AD or not. Along with the vision embeddings, we also input the length of the clip and the L2 distance of the current clip with the previous and next clip. We finetuned the classifier head for 2 epochs on MAD training data. We saw slightly better results compared to AutoAD-2. We present the results in the table below. Since AutoAD II code of temporal classification is not available, we present the best numbers from the corresponding paper for this model.
>
> Model | AUROC (higher better)
> | -------- | ------- |
> AutoAD II | 0.78
> Ours | 0.8
>
> We believe it is not possible to 100% accurately predict whether AD is needed in a given pause or not for shorter durations since it is quite subjective and we noticed differences even within movies in the eval-split (which are most likely annotated by different humans). We will add this additional analysis with exhaustive details to the camera-ready version.
>
> [1] - Liu, Hui, and Xiaojun Wan. "Models see hallucinations: Evaluating the factuality in video captioning." arXiv preprint arXiv:2303.02961 (2023).
>
> -------
> Thank you again for your thoughtful review. We hope we addressed your concerns/questions sufficiently and we would appreciate it if you would consider raising your score in light of our response. Please let us know if you have additional questions we can address.

---

> > ### Comment · Reviewer_suoE · 2024-08-12
> >
> > Thank you to the authors for the detailed response to my review! I will be updating and finalizing my rating after all discussion periods are complete, but this response is good overall and I believe strengthens the quality of the work. The temporal localization and additional discussions are much appreciated. Below, a quick response to one subset of the rebuttal above:
> >
> > *Re: analysis, verbosity, and hallucinations*: The analyses provided by the author are helpful to better contextualize the method.
> > - I recognize that there are not good hallucination metrics for open-ended video captions, but the character analysis is one that (at least in terms of the subject of the caption) does help to address this, especially given the movie domain. For general context, when the model does not mention the correct character, what are examples of the incorrect outputs that the model does provide instead? (also to confirm, are the pronoun substitutions aligned to the character identity?).
> > - It's unclear how well-correlated the hallucination scores (with the VLM "judge") are with human judgements (especially since VLMs can hallucinate themselves), but this analysis can still be useful. If the authors have additional examples that qualitative show the different scores that the model-based method outputs and can include that here/in their final supplement, that would be much appreciated.

---

> > > ### Author Response · Authors · 2024-08-13
> > > **Thank you Reviewer suoE!**
> > >
> > > We are glad to hear the reviewer is satisfied with our rebuttal. We address your queries below-
> > >
> > > > when the model does not mention the correct character, what are examples of the incorrect outputs that the model does provide instead? (also to confirm, are the pronoun substitutions aligned to the character identity?).
> > >
> > > We used Spacy for extracting Proper Nouns from each ground-truth AD to conduct character analysis. From the qualitative analysis of mistakes, we noticed that many a time, the character name is replaced with pronouns. We also observed a small percentage of instances where the model incorrectly refers to characters, for example, calling `Lisa` as `Susan` even though there is no character named `Susan` exists in the movie. This is perhaps a bias inherited from the training data or LLM itself.
> > >
> > > To quantify the correctness of predicted pronouns, we did not find any off-the-shelf tools to associate character names with pronouns. From qualitative analysis, the outputs seem reasonable most of the time. We will add this analysis, all the experiments done for rebuttal and brief discussion regarding the open problems in evaluations to the appendix of the paper.
> > >
> > > >  If the authors have additional examples that qualitative show the different scores that the model-based method outputs and can include that here/in their final supplement, that would be much appreciated.
> > >
> > > To give you a sense of scores, we provide some scenes from the analysis, CALVIN's predictions and Gemini's score in the table below. We see in current form VLM judge penalizes when a name is missing or the description diverges too far from the original description. We believe an in-depth analysis is needed to perfect the prompting of VLM-judge by looking at many diverse scenarios (which is a study on its own and too out of scope for this paper). We will add these examples and a discussion to the appendix of the camera-ready version of the paper.
> > >
> > > Ground Truth | Calvin prediction | VLM Judge score (lower better)
> > > --- | ----| ----|
> > > Swinging around, Lenihan aims his gun at the sky but sees nothing. | The soldier points the gun and looks at an apartment | 5
> > > Behind him, an alien emerges from the pool. | A robot is in the pool | 3
> > > Lenihan wheels around. | Lenihan turns | 1
> > >
> > >  Please let us know if you have any additional questions we can address.

---

> > > > ### Comment · Reviewer_suoE · 2024-08-13
> > > >
> > > > Thank you to the authors for the additional details and reply! Want to acknowledge that I have read this and will be incorporating into the final rating update after the next discussion phase.

---

### Official Review · Reviewer_jqA2 · 2024-07-19

**Soundness:** 3
**Presentation:** 3
**Contribution:** 3
**Rating:** 6
**Confidence:** 3

**Summary:**

Note: Raised score by 1 point after reading reviews, responses and the concerns addressed in the rebuttal phase.

----
This work introduces a video LLM model that can describe movie scenes in context incorporating names of characters and generate short contextual descriptions. They train a model on data from image question answering datasets and video description using context from previous frames, these enable their model to generate better contextual descriptions of events.
They evaluate their model on Movie Audio Description (MAD) and TV-Caption datasets and show improved performance.

**Strengths:**

* Their model tuning strategy of using context of the video is well motivated.
* Their tuned model shows good performance on both the MAD-eval dataset and TV-caption datasets.
* Ablations are explained clearly and evaluated well.

**Weaknesses:**

* With regard to evaluations, for movie audio description task, the CMD-train and and CMD-eval [1,2] datasets are also available. It’s not specified why the CMD dataset has not been used in this work, at least for eval.
* It seems valuable to have a baseline with the closed MultiModal LLMs e.g. GPT-4o and Gemini 1.5 Pro all of which claim to have superior video captioning capabilities.

**Questions:**

1. Have you evaluated on the CMD-eval dataset[1,2]? do you find the performance to be different? If not, is there a specific reason for not utilizing the CMD-AD eval dataset?
2. Do you evaluate on any closed multmimodal LLMs? If you tried, did you face any issues? If not, is there a reason not to try?

[1]Bain, Max, Arsha Nagrani, Andrew Brown, and Andrew Zisserman. "Condensed movies: Story based retrieval with contextual embeddings." In Proceedings of the Asian Conference on Computer Vision. 2020.
[2] https://www.robots.ox.ac.uk/~vgg/research/autoad/

**Limitations:**

Limitations are discussed at the end of the appendix. The reviewer’s understanding is that this woul dhave to move to the main paper. Please check the guidelines.

---

> ### Author Rebuttal · Authors · 2024-08-07
>
> Dear Reviewer jqA2, thank you for your positive review of our work, and your interest in our modeling strategy. There were a few questions raised regarding dataset choices and evaluation of API models that we're happy to answer below:
>
> **[Evaluation of the CMD dataset]**
> CMD is a classic video dataset, but we did not evaluate this dataset for several reasons. First, and most pragmatic, the download links to the raw data are no longer publicly available. Second, the CMD dataset’s descriptions are metadata from the YouTube clip, Wikipedia, and IMDb. While these descriptions give a general sense of the scene, they are not strictly scene audio descriptions. Finally, in the CMD paper, the authors evaluate models for retrieval tasks but not for text generation tasks. While we hold the dataset in high regard, it is not so clear to us how well performance on the CMD eval split measures the fine-grained AADs that we're looking for with modern models.
>
> However, similar to another related dataset CinePile[1], the CMD dataset might prove useful in the pre-training stage of the model.  While pretraining is a bit out of scope for us within the rebuttal phase, we will provide the results of the model variants of these two datasets in the mix for the camera-ready version.
>
> **[Evaluation of closed MultiModal LLM for the task]** Due to copyright reasons, the MAD dataset provides only CLIP embeddings but not the raw videos thus preventing us from evaluating closed MM-LLM models directly on MAD. This makes it impossible to test this dataset with general-purpose multi-modal APIs, which only accept video input (not CLIP embedding input). Nevertheless, we acquired scenes from 2 movies of the test set from YouTube - How Do You Know and Battle Los Angeles (approx 300 AD scenes) for this rebuttal and ran Gemini Pro. We present Google Gemini Pro and CALVIN’s results on this subset below. We chose Gemini Pro as a commercial model since it performs better than GPT-4o according to recent video understanding benchmarks[1,2].  While Gemini’s results are better, CALVIN’s numbers are close behind for a much smaller model, trained with research computing and public data. Since it is not known what is in its training mix, Gemini may have some level of memorization of the evaluated data while CALVIN never saw this data in training. We will add these results to the appendix in camera-ready.
>
>
> Model | BertScore |  CIDER |  ROUGE-L | SPICE
> | -------- | ------- |  -------- | ------- | ------- |
> Gemini | 43.23 | 28.11 | 18.54 | 8.99
> Calvin | 41.42 |  26.90 | 17.88 | 8.23
>
> [1] - Rawal, Ruchit, et al. "Cinepile: A long video question answering dataset and benchmark." arXiv preprint arXiv:2405.08813 (2024).
>
> ---------
> Overall, we appreciate the reviewer's insights, and these additional experiments have added depth to our analysis. We hope we addressed your questions sufficiently and we would be grateful if you would consider raising your score. Do you have any additional questions we can address?

---

> > ### Comment · Reviewer_jqA2 · 2024-08-12
> > **Satisfactorily addressed raised concerns**
> >
> > Thanks for the response and additional numbers! I also looked at the concerns raised by other reviewers, thanks for thoughtful responses to those reviews as well!
> >
> > It's unfortunate that there aren't other datasets to properly evaluate CALVIN against Multimodal LLMs. I definitely appreciate the  quick experiment with 2 movies that you were able to get hold of, and it is good to be able to put CALVIN's performance in perspective with these much larger and expensive off-the-shelf API models.
> >
> > Please include these discussions in the main paper or the supplement along with some notes on future direction of how one can evaluate models on this task. I will increase my score based on the response

---

> > > ### Author Response · Authors · 2024-08-13
> > > **Thank you!**
> > >
> > > We are glad to hear that you are satisfied with our response. We will include all the additional studies conducted for the rebuttal in the appendix of the paper.
> > >
> > > Additionally, we will add a discussion in the conclusion section addressing the limitations of the current MAD dataset and the need for a new 'raw video' dataset.

---

### Author Rebuttal · Authors · 2024-08-07

We would like to thank all four reviewers for their highly constructive feedback! We very much appreciate their assessment that our paper has “**thorough ablations**”[jqA2,suoE,Y2oh,GsDL], “**good performance**”[jqA2,suoE,Y2oh,GsDL], “well motivated”[jqA2], “important…novel problem”[suoE, GsDL], “well-written”[Y2oh].

The reviewers raised many excellent and thought-provoking questions and we conducted many experiments to address these questions. This interaction added more depth to the paper.

### [A brief summary of questions and additional results]
Here we summarize the additional analyses we conducted as part of the discussion. All the new analyses will be added to the appendix of the camera-ready version of the paper.
1. **Evaluation of closed MultiModal LLM for the task:** Since MAD dataset does not provide “raw” video data, we collected videos from YouTube for 2 movies from the test set and evaluated Gemini-Pro on the same. Results show Calvin has comparable performance (albeit slightly worse than Gemini) despite being a much smaller model. (For more details, see response to )
2. **Character analysis, verbosity, and hallucination aspects of captions:** For verbosity, we computed the average number of tokens of predictions (from Table 2). For character analysis, we computed the percentage of times the model can predict the proper nouns correctly. For hallucinations, we used the Gemini model as a judge to predict the level of hallucinations in the prediction against the ground truth. In all the metrics, our model Calvin did better than other open-source models. (For more details, see response to Reviewer suoE)
3. **Temporal localization of AD:** We trained a simple classifier head on top of Q-former and are able to get comparable results to AutoAD-II (For more details, see response to Reviewer suoE)
4. **How to do contextual captioning when ground truth context doesn’t exist?** Section 5 in the main paper is dedicated to this. Additional results show Calvin outperforms baselines in this scenario as well. (For more details, see response to Reviewer Y2oh)

Here are two points raised by couple of reviewers that we want to re-emphasize. We will add this discussion to the related work section in the camera-ready version.

### [Are baseline comparisons reasonable?]
We believe the comparisons with baselines are reasonable due to the following two factors:

1. The scale of the training data is the same, although the data mixtures differ, which is a core contribution of our paper. We also discussed in detail how to clean the data to improve performance, which was not addressed in previous works.
2. The size of the previous models and Calvin is the same. Videollama and Calvin have a similar number of parameters. Our improvements result from thorough experimentation and study of different components of the model pipeline.

### [How is Calvin different from previous methods?]
The first main difference between CALVIN and other models is how the video representations are projected onto the Language model space. While previous methods such as videoLLama used 2 projection Q-formers (with one component frozen and another trained from scratch) in sequence without any justification for this choice. We simplified the pipeline with a single Q-former that is trained from scratch. Merging of parameters into single module simplifies the architecture for thorough ablations, and we also found that the results are slightly better if there is 1 Qformer vs 2 Qformer blocks, perhaps due to slight reduction in number of parameters. The second main difference is the LLM finetuning and the usage of LoRA. Our experiments showed that unfreezing the LLM in stage 2 can lead to significant performance gains, while previous methods such as VideoLlama had that component frozen throughout the training.

Many recent works such as MM-1 [1], PaliGemma [2] have shown that while some critical architectural choices such as positional embeddings or certain activations could be important, most of the gains came from the data mixtures. Compared to previous works, we are also the first ones to propose the use of image VQA data in training and the conversion of caption data into synthetic VQA via LLMs and to show that adding these into training can improve performance.


-----
In conclusion, our work includes a number of key contributions concerning architectural choices for video LLM, model training, and dataset choice and setup. On the architectural side, we show that we can simplify/unify complicated input pipelines from prior work with a single Q-former, and we highlight the importance of continued training in the LLM part of the model.
We believe this can help video-understanding researchers make calls while training the next-gen models. For example, we are one of the few video-understanding papers that calls out the importance of data cleaning and data mixtures which is also echoed by recent works such as MM-1 [1], and PaliGemma [2]. We further introduce novel few-shot adaptations of the model for new movies which can make the model useful in practice and take it beyond the academic realm.

[1] - McKinzie, Brandon, et al. "Mm1: Methods, analysis & insights from multimodal llm pre-training." arXiv preprint arXiv:2403.09611 (2024).
[2] - Beyer, Lucas, et al. "PaliGemma: A versatile 3B VLM for transfer." arXiv preprint arXiv:2407.07726 (2024).

---

### Decision · Program_Chairs · 2024-09-25

**Decision:**

Accept (poster)

**Comment:**

This paper proposes a video LLM that uses previous movie context to generate contextual scene descriptions. Experiments on two datasets demonstrated the effectiveness of the approach.

Initially, the paper received four borderline accept scores. Reviewer jqA2 pointed out the lack of evaluation datasets especially the CMD dataset and no closed MLLMs as baselines. Reviewer suoE, Y2oh and GsDL all raised major concern on the novelty of the approach. Additionally, reviewer suoE questioned the evaluation metrics and reviewer Y2oh requested to align data for comparison and ablations w/o relying on groundtruth contexts.
The rebuttal provides more result comparisons, evaluations across different aspects and further analysis, which effectively addressed most of these concerns.

Based on this, three reviewers increased their scores to weak accept and one remained borderline accept. The AC thus recommends accepting the paper.
The authors should ensure that all promised revisions are included in the final version.